# Transfer Effects of a Multiple-Joint Isokinetic Eccentric Resistance Training Intervention to Nontraining-Specific Traditional Muscle Strength Measures [note 1]

**DOI:** 10.3390/sports11010009

**Published:** 2023-01-04

**Authors:** Steven Spencer, Brennan J. Thompson, Eadric Bressel, Talin Louder, David C. Harrell

**Affiliations:** 1Kinesiology and Health Science Department, Utah State University, Logan, UT 84322, USA; 2Movement Research Clinic, Sorenson Legacy Foundation Center for Clinical Excellence, Utah State University, Logan, UT 84322, USA

**Keywords:** training specificity, leg press, one-repetition maximum

## Abstract

Relatively few investigations have examined the transfer effects of multiple-joint isokinetic eccentric only (MJIE) resistance training on non-specific measures of muscle strength. This study investigated the transfer effects of a short-term MJIE leg press (Eccentron) resistance training program on several non-specific measures of lower-body strength. Fifteen participants performed Eccentron training three times/week for four weeks and were evaluated on training-specific Eccentron peak force (EccPF), nontraining-specific leg press DCER one-repetition maximum (LP 1 RM), and peak torques of the knee extensors during isokinetic eccentric (Ecc30), isokinetic concentric (Con150) and isometric (IsomPT) tasks before and after the training period. The training elicited a large improvement in EccPF (37.9%; Cohen’s d effect size [ES] = 0.86). A moderate transfer effect was observed on LP 1 RM gains (19.0%; ES = 0.48) with the magnitude of the strength improvement being about one-half that of EccPF. A small effect was observed on IsomPT and Ecc30 (ES = 0.29 and 0.20, respectively), however, pre-post changes of these measures were not significant. Con150 testing showed no effect (ES = 0.04). These results suggest a short term MJIE training program elicits a large strength improvement in training-specific measures, a moderate strength gain transfer effect to DCER concentric-based strength of a similar movement (i.e., LP 1 RM), and poor transfer to single-joint knee extension measures.

## 1. Introduction

Eccentric-based resistance training has received considerable attention from researchers and practitioners in recent years. As it is a common goal of strength training to increase functional capacity, it is important to measure the specific adaptations that occur in response to eccentric training modalities, as well as how well eccentric strength improvements induced by eccentric-based training methods transfer to other measures of muscular strength that are characteristically different than that conducted during training. At present this is especially true of eccentric training transferability characteristics since it has been an area less investigated in the context of specificity than more traditional forms of resistance training.

While investigations into the transferability of strength improvements following single-joint, isokinetic, eccentric-only (SJIE) resistance training programs have consistently observed significant increases in strength production capability in the training-specific mode, observations of the transferability of these gains to nontraining-specific contraction modes (specifically isokinetic concentric as well as isometric torque production) have yielded conflicting results [1,2,3,4,5,6,7,8,9]. Few studies have directly examined the effects of eccentric isokinetic training on isotonic one-repetition maximum strength (1 RM). Coratella et al. [7] observed that SJIE resistance training of the knee extensors increased isotonic 1 RM performed on a dynamic constant external resistance (DCER) leg extension machine by an average of 4.4 kg (effect size [ES] = 0.60). These observations suggest isokinetic eccentric exercise may have a moderate effect on isotonic 1 RM improvements. Further research is needed to determine if the strength gain transfer from isokinetic eccentric-only training to isotonic/DCER testing occurs in a multi-joint context.

Investigations examining eccentric training utilizing multiple-joint movements have increased in recent years [10,11,12,13,14]. Unfortunately, research on isokinetic eccentric-only multiple-joint training in the context of specificity is relatively sparse. Papadopoulos et al. [15] observed that eight weeks of twice-weekly isokinetic eccentric leg press training with relatively high loads (70–90% of maximum eccentric strength) induced large improvements in maximum force during isokinetic eccentric (64.9%) and isokinetic concentric (32.2%) leg press at the same contraction length, suggesting a moderate transfer effect to the concentric muscle action.

A limitation of the previously mentioned study is that the eccentric training groups were not evaluated on their isotonic strength (1 RM) in a movement that is similar in form (other than contraction type) to the multiple-joint eccentric leg press [15]. Given a multiple-joint model has more specificity to measures of function, such as walking, running, jumping, etc., it would be valuable to further investigate regarding its ability to transfer to a multitude of different strength measures. Further investigation of this area would elucidate the ways in which eccentric gains may be best utilized for developing an individual’s specific areas of weakness, and may be useful as a monitoring tool for providing insight on how a given magnitude of eccentric strength gains may relate to a given magnitude of gains across a number of other types of strength-based outcomes. Therefore, the purpose of this study was to determine the extent to which multiple-joint eccentric isokinetic leg press training-induced strength gains influence isotonic concentric leg press 1 RM and single-joint isometric and isokinetic peak torque (PT) of the knee extensors. We hypothesize that there will be significant transfer of muscular strength from the eccentric training mode to the DCER leg press movement, and poor transfer to single-joint isokinetic and isometric movements.

## 2. Materials and Methods

### 2.1. Subjects

Fifteen college-aged men and women volunteered to participate in the study. The demographics of the study sample are as follows: *n* = 6 females, 9 males; mean ± SD: age = 22.6 ± 2.0 years, height = 176.0 ± 7.7 cm, mass = 73.3 ± 18.2 kg. Eligibility criteria to participate in the study required the participants to be between the ages of 18–30 years. Participants were required to be informally classified as recreationally active, such that they were allowed to be involved in sports and moderate dose physical activity but were not allowed to be regularly engaging in resistance training (< three times in the previous month) nor could they be involved in aerobic exercise (jogging, aerobics) for more than 30 min per day, five days per week. Participants must not have had any lower limb injuries nor had surgery on the lower limbs within one year prior to the beginning of the study. Participants were required not to consume nutritional supplements for muscle growth (i.e., Creatine) during the study nor three months prior and were additionally not allowed to consume nonsteroidal anti-inflammatory drugs (NSAIDs) during the study period. Participants were instructed, and repeatedly encouraged to keep their dietary intake as consistent as possible during the study.

The study was approved by the Utah State University Institutional Review Board and all participants read and signed an informed consent document prior to study participation.

### 2.2. Procedures

This study utilized a repeated measures (pretest/posttest) design to test the hypotheses following a 4-week training intervention. The present study uses a subset of data from a larger investigation [11]. Specifically, the training protocol was used, and the Eccentron maximal strength data was reported in the previous study, but was not analyzed in the context of the specificity question per the current study. This particular variable is reported again in the present study, but is being used to compare with new (not published before) strength measures (e.g., leg press 1 RM and all Biodex data) for the benefit of assessing and comparing a range of different muscle strength measures. Upon enrollment, participants completed one practice session to become familiarized with all performance testing procedures prior to the formal pretest in an attempt to minimize the influence of learning on testing outcomes. All testing was performed at the same time of day (±2 h), and always occurred in the order as presented below. Posttesting occurred 4–6 days following the last training session to allow for full recovery. All testing and training sessions were preceded by a brief warm-up consisting of cycling on an ergometer at 50 watts for two minutes, followed by a brief dynamic stretching routine which was described in the previously published report [11]. All exercise training and testing was closely supervised by experienced research investigators.

### 2.3. Outcome Measures

#### 2.3.1. Biodex

Participants were tested for single-joint isokinetic and isometric strength capacities of the knee extensor muscle group on an isokinetic dynamometer (Biodex System 3, Biodex Medical Systems, Shirley, NY, USA) and followed the procedures previously reported by Gordon et al. [10].

The participant was seated on the Biodex and the restraining straps were placed over the waist, chest, and thigh. The seat was adjusted such that the rotational axis of the knee joint was aligned with the rotational axis of the dynamometer. This seat position was recorded and used for the posttest for each individual. The arm of the Biodex was secured to the lower leg approximately five cm above the malleolus. Participants performed a brief warm-up of ten isokinetic knee extension and flexion repetitions at 120°·sec^−1^ with 75% of maximum effort.

Participants then performed two maximum voluntary isometric contractions (MVICs) of the knee extensors at a joint angle of 60° below the horizontal plane [16], with a one-minute rest between MVICs. This was followed by the isokinetic testing that was based on a previously established testing protocol [10,17]. For this, participants performed three maximum isokinetic concentric knee extensor contractions at 150°·sec^−1^. These were followed by two maximum isokinetic eccentric knee extensor contractions at 30°·sec^−1^ with a two-minute rest between each attempt.

The Biodex dynamometer was configured with a Biopac data acquisition system (MP150, Biopac Systems Inc., Santa Barbara, CA, USA) which sampled the torque signal at 2000 Hz. Custom written software (LabVIEW 2016, National Instruments, Austin, TX, USA) was used to process the data following the methods of Gordon et al. [10]. Briefly, the voltage signals were digitally converted and scaled to torque units (Nm), and filtered using a zero phase shift, fourth-order Butterworth filter with a 50-Hz low-pass cut-off frequency.

The isometric signal was gravity-corrected by subtracting the baseline value of the participant’s limb weight from the entire torque signal. Isometric PT (IsomPT) was quantified as the highest 500-ms epoch during the plateau phase of the MVIC. The isokinetic torque signal was also gravity-corrected for limb weight in accordance with the procedures of Aagard et al. [18]. The isokinetic concentric PT (Con150) and eccentric PT (Ecc30) were calculated as the mean value of the highest 25-ms epoch of the torque-time signal and the highest contraction was used for data analysis.

#### 2.3.2. Eccentron Strength Test

Participants were tested for their maximum eccentric strength (peak force, N) on a multiple-joint, isokinetic eccentric dynamometer (Eccentron, BTE Technologies Inc., Hanover, MD, USA). Reliability data assessed from our lab on 11 healthy subjects has demonstrated this test has high reliability per the following reliability statistics: intraclass correlation coefficients = 0.94, standard error of measurement = 146.28 N, and minimum difference to be considered real = 405.472 N. These testing procedures have been reported in our previous study [11]. Briefly, participants were seated on the machine with the seat adjusted per the manufacturer guidelines so that the knee joint was set to an angle of 30° when fully extended. During the testing, the pedals moved toward the participant in an alternating motion, so that each leg worked isolaterally in an alternating, repetitive manner. The speed of this motion was set at 23 cycles per minute (a medium velocity for this movement). Testing of the Eccentron at this speed showed a movement time of 1.40 s for each repetition, with an average velocity of 0.157 m/s. The testing consisted of a total of 12 maximum effort repetitions, six for each leg. Participants were instructed to maximally resist the motion of the pedal as it moved towards them, then relax that leg as it moved away, at which point they pushed maximally with the other leg. Participants were given a familiarization of the testing protocol ~72 h prior to the testing session. During the testing period, participants received verbal encouragement to reinforce maximum effort.

#### 2.3.3. Leg Press Strength Test

Participants were assessed for lower-body dynamic strength utilizing a traditional leg press machine set at a 45° angle and followed the NCSA guidelines for 1 RM testing. Briefly, the participant was instructed to warm-up with a light resistance that easily allowed the completion of 5–10 repetitions, after which they rested for one-minute. For the following set, 10–20% of the initial warm-up weight was added to create a weight that could be lifted 3–5 times, followed by a two-minute rest period. The next set added an additional 10–20%, to produce a weight that could be lifted 2–3 times followed by a 2–4 min rest period. The 1 RM attempts followed, with the default condition of adding 10–20% additional weight with each successful attempt or based on participant preference. Each 1 RM test was followed by a 2–4 min rest period [19].

### 2.4. Eccentric Training Program

The eccentric training has been described previously [11]. Briefly, the motor-driven eccentric isokinetic dynamometer (Eccentron) was used as the multiple-joint training modality. Prior to all training sessions, a brief warm-up was performed consisting of cycling on an ergometer at 50 watts for two minutes, followed by a brief dynamic stretching routine which was described in the previously published report [11]. For the Eccentron training, the manufacturer-designed protocol consists of a one-minute warm-up at half of the workout target force, followed by the workout period, then a one-minute cool down period also at half of the workout target force. The participants completed a two-minute workout phase (excluding the one-minute warm-up and cool down phases) three times per week on non-consecutive training days. The training velocity was set at 23 cycles per minute for all training, and this may be considered as a moderate velocity for this dynamometer.

During the first week of the study, two relatively light sessions were performed with a target force of 45 and 50% maximum effort for sessions 1 and 2, respectively, to allow for familiarization to the movement and to mitigate extreme soreness levels.

The training load progression has been reported for the study [11] and was increased incrementally throughout the duration of the training program. Briefly, the intensity progression was based on the percentage of baseline maximum eccentric strength and was as follows: week 1 = 50, 52.5, and 55%; week 2 = 60, 62.5, and 65%; week 3 = 70, 72.5, and 75%; week 4 = 75% for all three sessions.

### 2.5. Statistical Analyses

Dependent t-tests were used to evaluate the effects of the training program (pretest vs. posttest) on all dependent variables. The Cohen’s d ES statistic was calculated to evaluate the meaningfulness of the training effects, with values of 0.2, 0.5, and 0.8 being considered as small, medium, and large effect sizes, respectively [20]. Pretest to posttest relative change (%) scores were calculated for each participant using the means for each dependent variable and assessed for normal distribution using the Kolmogorov–Smirnov test of normality. Pearson correlation coefficients (*r*) were used to assess the relationship between the change scores of the dependent variables unless Kolmogorov–Smirnov results indicated that normality was violated, in which case relationships between the change scores were assessed using Spearman’s rho correlation coefficient. SPSS software (version 25; IBM SPSS, Inc., Chicago, IL, USA) was used for all statistical analyses. An alpha level of *p* ≤ 0.05 was used to determine statistical significance.

## 3. Results

Posteriori power was calculated using the Eccentron ES value of 0.86 for a two tailed, dependent t-test, and the resulting statistical power was 0.81. All subjects completed all training and testing sessions during the 4-week intervention. However, two subjects’ posttest eccentric strength exceeded 3338 N, which is the upper limit capability of the Eccentron, thus, the final sample size was *n* = 13 for the testing data after removing these two subjects’ data from the analyses.

Means and standard deviations (SD), relative change scores, and ES values for all variables are presented in Table 1. For the Eccentron peak force, there was a training effect showing a significant increase (*p* < 0.001) and the Cohen’s d ES was large (0.86). The leg press measure also presented a significant increase (*p* < 0.001) and the ES was moderate (0.48). All single-joint variables including IsomPT, Ecc30, and Con150 did not show significant improvements (*p* = 0.16–0.81), but the ES values for IsomPT and EccPT30 suggest a small training effect (ES = 0.29 and 0.20, respectively) was present. Con150 was the only variable to show no training effect based on the Cohen’s d ES (0.04). Figure 1 shows the mean relative change scores across all muscle strength variables for each participant. Appendix A present the data stratified by sex.

Correlations between all muscle strength measures for the pretest to posttest relative change scores are presented in Table 2. The change score distribution for the Ecc30 variable was observed to be the only variable to violate normality (Kolmogorov–Smirnov *p* < 0.001), therefore correlations for this variable are reported as Spearman’s rho. All other variables are reported as Pearson’s *r*. Analyses revealed the correlations between the Eccentron peak force change scores and the other muscle function measures were significant for IsomPT (*p* = 0.04), approached significance for leg press 1 RM (*p* = 0.055) and Con150 (*p* = 0.057) and was not significant for Ecc30 (*p* = 0.16).

## 4. Discussion

The hypothesis of the study was largely fulfilled such that the muscular strength gain transfer from the Eccentron training was rather large to the DCER leg press task, but minimal to the single-joint isokinetic and isometric tasks.

An important finding of the present study was the significant increase observed in the DCER leg press 1 RM measure, which showed a mean improvement of 19.0%. Comparatively, the mean improvement in DCER leg press 1 RM (19%) was about half of that observed in peak force on the Eccentron (37.9%). This finding seems to support previous observations from Papadopoulos et al. [15], who observed an improvement of 64.9% in isokinetic eccentric leg press strength and a 32.2% improvement in isokinetic concentric leg press strength following 8 weeks of isokinetic eccentric leg press training. These observations suggest that multiple-joint isokinetic eccentric-only exercise has a moderate effect on strength gains of a similar movement pattern that is concentric-based, such that the magnitude of concentric DCER gains can be expected to be approximately half of what the training-specific gains are for this type of training program.

The single-joint testing of the knee extensors displayed overall low transferability across the three variables. There were no significant pre-post changes observed in IsomPT, Ecc30 or Con150 (*p* = 0.16, 0.33 and 0.81, respectively). However, ES values suggest that there was a small training effect for IsomPT (ES = 0.29) and Ecc30 (ES = 0.20), but no effect for Con150 (ES = 0.04). These results suggest that the training effect from the Eccentron to single-joint measures is quite negligible. The lack of specificity in terms of both the joints recruited and movement pattern differences between the multiple-joint leg press movement and the single-joint knee extension movement likely explained this poor transfer outcome. During the Eccentron workouts, participants anecdotally reported feeling the resistance mostly in the hip extensors, specifically the glutes, as well as reporting the greatest amount of soreness in the same muscle group. The movement pattern is also vastly different between these movements such that the leg press incorporates a closed kinetic chain pressing movement, whereas the isokinetic knee extension incorporates an open kinetic chain and widely angular pattern. Based on our observations, it seems that single-joint measurements of the knee extensors are a poor measurement of muscle strength changes when this training modality (multiple-joint, eccentric) is utilized in an active young adult population.

Contrasting the observations of the current investigation, Kim et al. [21] observed substantially improved performance in single-joint isometric knee extension (ES = 3.43), isokinetic knee extension at 60°·sec^−1^ (ES = 1.71), and a 10-s test of power in the knee extensors and flexors (ES = 0.72) following an eight week Eccentron training routine with older adults. These results showed the Eccentron training induced a large improvement in these measures despite the lack of training-specificity including the joint and movement pattern discrepancies as noted above. The diverging results may stem in part from the differences between the methods and training procedures. The testing and training parameters performed in the Kim et al. [21] study differed from those of the current study in that Eccentron strength was not reported for either pre- or posttesting (it was measured only during pretesting for the purpose of resistance dosing for the training) and so the changes in the training-specific multiple-joint eccentric strength are not available for comparison with the nontraining-specific variables. Additionally, for the training Eccentron loading was set at 50% of pretest strength for the duration of the training period versus beginning at 50% for the first week and progressing to 75% in the present study, and the training sessions were performed twice a week compared to three times a week in this study. Additionally, and perhaps most importantly, their training sessions were overall much longer in that they were 20 min each session for week one (at 18 reps/minute) and increased to 30 min each session during weeks 2–8 (at 23 reps/minute) with two, 2 min rest periods and their program duration was eight weeks compared to four weeks in this study. Overall the training program implemented by Kim et al. [21] had a much higher training dose compared to the present study, given that the sessions were at least three to five times longer and their training duration was twice as long. These differences in training period length and volume would likely have contributed to the larger nontraining-specific strength testing gains observed in the Kim et al. [21] study. Unfortunately, their study did not report on the Eccentron strength variable, and so it is not possible to make direct comparisons between the training-specific eccentric strength gains and the nontraining-specific gains among the variables assessed in their study with the present study’s specificity values. Finally, it is likely the case that the older adults in the Kim et al. [21] study were relatively weaker than the active young adults in the current study, and therefore had greater potential for overall strength gains, which were able to be more easily and effectively transferred and displayed in the non-specific strength testing measures. Some support for the effect of their study population’s propensity for gains may be seen in their data, as the large effect sizes observed in the Kim et al. [21] study were partially the result of small standard deviations in the outcome measures which reflects the low variability or high consistency in the gains among the study subjects.

Analysis of the correlation between change scores of the dependent variables revealed a significant correlation between the training-specific Eccentron peak force and nontraining-specific IsomPT (*p* = 0.04; *r* = 0.57), as well as correlations with leg press 1 RM (*p* = 0.055; *r* = 0.54) and Con150 (*p* = 0.057; *r* = 0.54) that approached significance. The change scores of Eccentron peak force were not significantly correlated with those of Ecc30 (*p* = 0.41; *r* = 0.41).

The relationships in the change scores between the training-specific variable (Eccentron peak force) and non-specific variables of leg press 1 RM, IsomPT, and Con150 yielded a remarkably similar level of correlation (*r*^2^ = 29.2–32.3), indicating approximately a third of the variance in Eccentron peak force changes explained the changes in these nontraining-specific variables. This is an interesting finding given the magnitude of change among these three non-specific measures were considerably different, and since testing on the Con150 variable did not indicate any improvement occurred. In light of these observations, it is worth noting that correlations do not assess differences among variables, only how one variable moves (i.e., increases or decreases) in comparison to another variable’s movement for each participant. A close examination of the data as shown in Figure 1 seems to support this finding. For example, the participants that showed large gains on the Eccentron peak force variable (see participants 4, 5, and 8) also tended to show relatively large gains on the leg press 1 RM, IsomPT, and Con150 variables, even though the magnitude of these gains was lower, to varying degrees, than the Eccentron variable. For participants who showed the lowest gains in Eccentron peak force, the changes in these same variables are, in general, relatively small (see participants 3 and 7). As a result, gains in Eccentron peak force were related to relative gains in some of the non-specific variables, even though the magnitude of these gains differed dramatically. On a per subject basis, change scores for IsomPT and Con150 tended to move similarly to Eccentron peak force change scores, but the magnitude of the change was drastically lower across the study sample.

There were some notable limitations of this study. As this analysis was done on a subset of a larger study there was a relatively small sample size for this study. Of this sample, two participant’s results had to be omitted from the statistical analysis, as their eccentric strength during posttesting exceeded 3338 N, the maximal force the Eccentron machine is capable of measuring. When this upper threshold is reached the pedals stop and no valid measurement can be observed. Additionally, the proprietary Eccentron strength testing protocol does not record the maximum force produced by the participant on the 12 repetitions of the testing, but instead eliminates the repetition with the highest force value for each leg and then takes the next highest value (so usually the third or fourth highest repetition is what is reported). Although, this way of determining maximum eccentric strength is likely appropriate for tracking changes in maximum eccentric force production capability over time, it does not report the actual highest eccentric value out of the multitude of maximum repetitions performed, which was the way the other variables were assessed (i.e., as the highest measure of several attempts). However, the difference between the highest and third or fourth highest repetition in a 12 maximal repetition test is quite miniscule.

The results of this study add to accumulating empirical observations on the potential benefits of eccentric-based resistance training. Previous research on this topic has observed that eccentric-based resistance training may yield substantial results on muscle mass and strength [8,22,23] with a lower rating of perceived exertion [24] and metabolic demand [25] compared to concentric-only or traditional isotonic resistance training. These factors may make eccentric overload training an attractive training model for both athletic and clinical populations because it may provide the opportunity for substantial gains to be achieved in a more tolerable and time-efficient manner, and it may be especially well-suited for those who may have a limited capacity to perform traditional forms of resistance training. While the benefits of eccentric training carry high relevance to athletic performance, these adaptations may benefit non-athletic populations as well. Kim et al. [21] observed increases in testing scores on chair sit-stand, gait speed and stair climbing in elderly adults (aged 72.38 ± 2.62 years) following an 8 week training program on an Eccentron. Notably, the increases in stair climb and gait speed were significantly greater in the Eccentron training group compared to a group that performed a conventional DCER training program [21]. Such findings indicate the potential for this form of training in clinical-based settings.

However, it should be noted that eccentric exercise, such as implemented in this research study, carries several disadvantages. First, the high load condition of eccentric overload training induces a significant amount of delayed-onset muscle soreness (DOMS), likely as a result of exercise-induced muscle damage [26], as well as the potential for joint pain which may result in cessation of training as observed in our previous study [10]. This process results in a temporary decrease in performance and is especially evident when the eccentric exercise being performed is novel to the participant [26]. However, during long-term training programs, these effects may be mitigated by the repeated bout effect, a phenomenon in which subsequent bouts of eccentric exercise results in less muscular damage and soreness in subsequent bouts [27,28,29]. Additionally, the training performed consisted of eccentric-only work and so there was no utilization (e.g., training) of the stretch-shortening cycle, which may limit the potential transfer of eccentric strength improvements to measures of stretch-shortening cycle function such as sprinting and jumping [10]. Furthermore, the training program utilized a unique training apparatus (Eccentron), which is unlikely to be found in a typical training facility.

## 5. Conclusions

A 4-week multiple-joint eccentric-only resistance training program elicited significant increases in performance on the training-specific outcome measure (Eccentron peak force), as well as on DCER 1 RM strength measured on a biomechanically similar movement (leg press). The observed strength gain in the leg press 1 RM was about half that of the Eccentron strength gain, suggesting a moderate transfer effect from an eccentric-only training protocol to a DCER concentric-based 1 RM movement that reflects a similar movement pattern that mostly only differs in type of contraction (eccentric- vs. concentric-based). There was generally a poor transfer of Eccentron performance improvements to isolated (single-joint) strength measurements of the knee extensors on the Biodex, suggesting that these types of measurements are a poor indicator of muscle strength improvements resulting from a multiple-joint isokinetic eccentric training program. The present observations may be of particular interest to professionals who work in clinical, rehabilitation or performance settings, where the potential for large strength improvements from short duration training sessions make eccentric exercise a particularly useful exercise modality for training-specific strength improvements. However, the transfer of these improvements to assessments of various nontraining-specific strength tasks is, to varying degrees, limited. Practitioners may consider the present observations to inform testing protocols for research designs that implement multiple-joint lower body eccentric training with the intent to most effectively capture the training gains in their designated outcome measures. In particular, when using single-joint isokinetic (Biodex) assessments, these observations suggest very large improvements in multiple joint eccentric-based strength are necessary to produce minimal changes in these parameters, which is most likely due to specificity factors.

When applying eccentric-based training programs, practitioners should consider biomechanical and contraction-type specificity when selecting methods for training and testing outcomes. The outcomes of this research suggest that for assessment of the efficacy of multiple-joint eccentric-only training methods, concentric-based testing (e.g., DCER 1 RM) of biomechanically similar movements are a moderately effective measure, while single-joint measures are relatively poor indicators of improvement.

## Figures and Tables

**Figure 1 sports-11-00009-f001:**
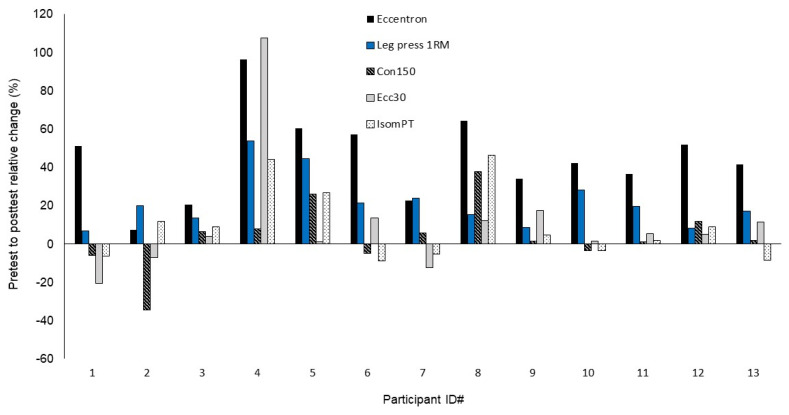
Pretest to posttest individual relative change scores for all muscle strength variables.

**Table 1 sports-11-00009-t001:** Mean (SD), change scores, *p*-values, and Cohen’s d effect size values for muscle function variables before (Pre) and after (Post) the training period.

Action	Variable	Pre	Post	Change Score (%)	*p*-Value	Cohen’s d
Eccentron	Peak Force (N)	1364.58 (599.23)	1882.28 (600.58)	37.94	*p* < 0.001	0.86
Leg Press	1 RM (N)	1909.31 (730.12)	2272.01 (793.18)	19.00	*p* < 0.001	0.48
Knee Extensors (Biodex)	IsomPT (Nm)	171.19 (42.30)	182.16 (33.65)	6.41	*p* = 0.16	0.29
	Ecc30 (Nm)	216.65 (59.89)	227.14 (44.29)	4.85	*p* = 0.33	0.20
	Con150 (Nm)	128.47 (40.73)	130.08 (37.66)	1.25	*p* = 0.81	0.04

Note: Cohen’s d values compare within-group pretest and posttest differences and are identified as being small, moderate, and large on the basis of values of 0.2, 0.5, or 0.8 respectively. Ecc30 = Biodex knee extensor eccentric muscle action at 30°·sec^−1^; Eccentron = maximal multiple-joint eccentric force on the Eccentron device; IsomPT = isometric PT of the knee extensors on the Biodex; Con150, Biodex knee extensor concentric muscle action at 150°·sec·^−1^.

**Table 2 sports-11-00009-t002:** Correlation matrix for the pretest to posttest relative change scores for all the muscle strength variables. Correlations are reported as Pearson’s *r* with the exception of the Ecc30 variable, which are reported as Spearman’s rho.

Variable	EccPF	LP 1 RM	IsomPT	Ecc30	Con150
EccPF		0.54a	0.57 *	0.41	0.54b
LP 1 RM			0.05	0.07	0.18
IsomPT				0.17	0.56 *
Ecc30					0.18
Con150					

All correlations are reported as Pearson’s *r* with the exception of the Ecc30 variable, which are reported as Spearman’s rho (see text). EccPF = Eccentron peak force; LP 1 RM = leg press one-repetition maximum; IsomPT = isometric peak torque (Biodex); Ecc30 = eccentric peak torque at 30°·sec^−1^ (Biodex); Con150 = concentric peak torque at 150°·sec^−1^ (Biodex). * *p* < 0.05, a *p* = 0.055, b *p* = 0.057.

## Data Availability

The raw data supporting the conclusion of this article will be made available by the authors upon request.

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
