# Peer review of "Transfer Effects of a Multiple-Joint Isokinetic Eccentric Resistance Training Intervention to Nontraining-Specific Traditional Muscle Strength Measures†"

_sports, 2023, doi:10.3390/sports11010009_

Round 1

Reviewer 1 Report

The manuscript titled "Transfer Effects of a Multiple-Joint Isokinetic Eccentric Training Intervention to Nontraining-Specific Traditional Muscle Function Measures" is interesting, I believe it is important to assist sports professionals in prescribing strength training. However the manuscript can be improved, for that, I have the following suggestions:

1) I suggest inserting a visual abstract to summarize the main findings of the study.

1.1) The introduction is good, it adequately presents the state of the art, however it is too long, I suggest reducing the introduction to 4 or 5 paragraphs including the justification, hypothesis, and objective.

2) Methods: Did the authors perform an a priori sample calculation? If yes, you must insert it in the methods. If not, they should state the reason in the methods, then they should calculate the a posteriori sample power for all the variables analyzed and state it in the results section or in a supplementary file.
2.1) Statistical analyses: The authors mention they used a repeated measures group design and this leaves the interpretation confusing, I suggest they remove that sentence. Inform that the pre and post comparisons were made using the t-test for paired samples. In addition, inform which variables showed a non-normal distribution and if you used non-parametric tests or transformed the data for statistical treatment. The appropriate data normality test for the study sample is the Shapiro-Wilk test, not the Kolmogorov-Smirnov test. I suggest they check the coefficient of variation together with the normality of the data, if the coefficient of variation is higher than 15% it justifies the use of non-parametric analyses. Finally, the authors should present pre and post analyses for the total sample and for the male and female samples individually.

3) Results: Should be restated by means of the requested adjustments to the analyses (see comment 2.1 above), I suggest that the authors use tables.

4) Discussion: The first paragraph begins by summarizing the results, this is not adequate for the session, I suggest you adjust it, mention only the objective and the initial hypothesis and inform if the hypothesis was fulfilled (totally or partially).

4.1) Discussion: remove all repetition of results! The discussion should be direct and objective and try to justify the findings with arguments based on sports training, exercise physiology, biomechanics, etc.

5) Conclusion: It should be objective and focused on answering the objective of the study.

Reviewer 2 Report

The purpose of this study was to determine the extent to which multiple-joint eccentric isokinetic leg press training-induced strength gains influence isotonic leg press 1RM, single-joint isometric and isokinetic PT of the knee extensors. 

In spite of being an article with data obtained from another larger project, the research is methodologically well presented. Although the writing provides sufficient information and correctly contextualizes the subject, especially the introduction, it is too long and should be summarized in a few paragraphs.

Here are my contributions: 

Introduction:

- In the second paragraph, reference is made to the advantages of eccentric training, but not to the disadvantages. Include the disadvantages as well.

- The introduction talks about the benefits of eccentric training for the athletic population but the article is focused on the active, non-athletic population. Clarify in that paragraph the possible benefits of eccentric training in the non-athletic population.

- The introduction is very extensive. It should not be a review of the subject, it should introduce and justify the review performed. For example, the third, fourth, fifth, seventh and eighth paragraphs are too specific and should be reduced. Many of the things discussed come up again in the discussion.

Methods:

- Participants were encouraged to maintain as consistent a diet as possible during the study, but were they given some sort of guideline to ensure minimum protein amounts in all of them?

  • Prior to the assessments/tests, did the subjects perform any kind of warm-up? Warm-up is only mentioned in the Leg Press Strength Test and Eccentric Training Program.

  • It is recommended that bicycle training is not performed at an absolute intensity (50 watts) but at a relative intensity (1 watt/kg for example), otherwise each subject will be warming up at a different intensity.

Discussion:

- Is eccentric training recommended for all populations despite the possible increased risk of injury in its execution?

Round 2

Reviewer 1 Report

in my conception the manuscript is adequate and ready for publication!

Reviewer 2 Report

The authors have adequately addressed the contributions made.